# Genome-Wide Identification, Evolution, and Expression Analysis of the *DIR* Gene Family in *Schima superba*

**DOI:** 10.3390/ijms25137467

**Published:** 2024-07-07

**Authors:** Changya Chen, Yanling Cai, Boxiang He, Qian Zhang, Dongcheng Liang, Yingli Wang, Hongpeng Chen, Jun Yao

**Affiliations:** 1School of Traditional Chinese Medicine, Guangdong Pharmaceutical University, Guangzhou 510006, China; 2Guangdong Provincial Key Laboratory of Silviculture Protection and Utilization, Guangdong Academy of Forestry, Guangzhou 510520, China

**Keywords:** *Schima superba*, dirigent family, abiotic stresses, gene expression analysis, genome-wide analysis

## Abstract

*Schima superba*, commonly known as the Chinese guger tree, is highly adaptable and tolerant of poor soil conditions. It is one of the primary species forming the evergreen broad-leaved forests in southern China. Dirigent proteins (DIRs) play crucial roles in the synthesis of plant lignin and lignans, secondary metabolism, and response to adversity stress. However, research on the *DIR* gene family in *S. superba* is currently limited. This study identified 24 *SsDIR* genes, categorizing them into three subfamilies. These genes are unevenly distributed across 13 chromosomes, with 83% being intronless. Collinearity analysis indicated that tandem duplication played a more significant role in the expansion of the gene family compared to segmental duplication. Additionally, we analyzed the expression patterns of *SsDIRs* in different tissues of *S. superba*. The *SsDIR* genes exhibited distinct expression patterns across various tissues, with most being specifically expressed in the roots. Further screening identified *SsDIR* genes that may regulate drought stress, with many showing differential expression under drought stress conditions. In the promoter regions of *SsDIR*s, various cis-regulatory elements involved in developmental regulation, hormone response, and stress response were identified, which may be closely related to their diverse regulatory functions. This study will contribute to the further functional identification of *SsDIR* genes, providing insights into the biosynthetic pathways of lignin and lignans and the mechanisms of plant stress resistance.

## 1. Introduction

*Schima superba* is one of the main tree species in the evergreen broad-leaved forests of southern China. Its stem bark is known for its efficacy in clearing heat and detoxifying the body, as well as its activity in inhibiting the growth of tumor cells [1]. It is widely distributed in most subtropical regions of China, exhibiting strong adaptability and tolerance to poor soil conditions. It is also an important native broad-leaved and timber species. With the continuous increase in demand and quality requirements for timber in China, *S. superba* is increasingly valued in various provinces in the south. To cope with severe biological and abiotic stresses, plants have spontaneously evolved a series of defense mechanisms to rapidly respond to complex environments and minimize damage. 

Abiotic stress is a common environmental factor affecting plant growth and development. Being sessile, plants face various challenges during their growth due to prolonged exposure to both biotic and abiotic stresses, leading to potential yield losses [2,3]. To counter these threats, plants have evolved multiple mechanisms that enable them to rapidly respond to complex environmental changes, thereby minimizing losses and efficiently utilizing resources to promote growth and reproduction [4,5]. Plants respond to biotic and abiotic stresses by activating various genes, including the Dirigent protein (DIR) genes [6,7]. *DIR* genes are involved in the synthesis of lignin and lignans, which can enhance the stress resistance of various crops [7]. They play a crucial role in plant defense against both biotic and abiotic stresses. The mechanical support and water transport provided by xylem vessels limit the entry of harmful environmental factors and enhance resistance to microbial pathogens and pests [8,9,10,11]. The *DIR* genes can regulate the biosynthesis of lignin and lignans, indicating that the DIR genes actively respond to both biotic and abiotic stresses.

The *DIR* proteins were first identified in *Forsythia intermedia* in 1997 [12]. The classical *DIR* gene structure has no introns, with most of the protein sequences containing only a single dirigent conserved structural domain and a few members containing two tandem dirigent structural domains [13]. *DIR* proteins are widely distributed in almost all vascular plants, including ferns, gymnosperms, and angiosperms [6,14,15]. Currently, *DIR* gene family members have been identified in multiple species, including *Arabidopsis thaliana* [16], *Oryza sativa* [17], *Cajanus cajan* L. [18], *Acorus tatarinowii* [19], *Populus trichocarpa* [20], *Setaria italica* [21], and *Solanum tuberosum* [22]. Previous studies have shown that the *DIR* gene family can be divided into five subfamilies, DIR-a, DIR-b, DIR-c, DIR-d, and DIR-e, but with the addition of more protein members in the *DIR* family, two new subfamilies, DIR-f and DIR-g, have emerged; the DIR-b and DIR-d subfamilies are combined together as the DIR-b/d subfamily [7,13]. Research has indicated that members of the DIR-a subfamily primarily participate in the stereoselective coupling reactions of lignans and lignins [13,23,24,25,26,27]. However, there are fewer reports on the directional functions of other subfamily members. 

*DIR* proteins regulate plant stress responses by participating in the control of metabolic synthesis pathways of lignans and lignin. A substantial body of research indicates that *DIR* proteins play a significant role in plant stress responses. For example, in the resistance to powdery mildew in *Cucurbitaceae* crops such as watermelon, muskmelon, and cucumber [28], the expression levels of the *ClDIR5*, *ClDIR6*, *ClDIR8*, and *ClDIR9* genes increase with the duration of inoculation, indicating the crucial role of the *DIR* gene family in plant resistance to abiotic stressors. The cucumber *CsDIR16* can increase the soluble sugar content and enhance POD enzyme activity under propamocarb stress, thereby mitigating the damage caused by reactive oxygen species to the plasma membrane [29]. Overexpression of *PtDIR11* in poplar leads to an increased lignin content and enhances resistance to the pathogen Septotis populiperda by regulating JA- and ET-related genes [30]. Overexpression of *AtsDIR23* in *Acorus tatarinowii* leads to an increased lignin content. *AtsMYB91* negatively regulates the expression of *AtsDIR23* by binding to its promoter, thereby regulating lignan synthesis [19]. *LrWRKY11* regulates *LrDIR1* expression through the salicylic acid(SA)/jasmonic acid (JA) signaling pathways, promoting lignin and lignan accumulation, and thereby enhancing Lilium regale resistance to Fusarium oxysporum [31]. *PnDIR* in Panax notoginseng participates in lignin and lignan biosynthesis and significantly increases resistance to Fusarium solani [32]. The overexpression of the *GhDIR1* gene in cotton leads to an increase in the lignin content, thereby enhancing the cotton’s tolerance to wilt disease [33]. The overexpression of the *GmDIR22* gene in soybeans can increase the total lignin content, thereby enhancing the plant’s resistance to soybean Phytophthora [34], and overexpression of *TaDIR13* can lead to an increase in lignin content in tobacco, enhancing its resistance to *Pseudomonas syringae* [35]. *FvDIR13* can promote lignin synthesis and play a role in disease resistance by regulating methyl jasmonate (MeJA) and SA [36]. In addition to the biotic stress response, *DIR* genes also participate in the growth, development, and abiotic stress response of plants. For example, *BhDIR1* shows significantly enhanced transcription under dehydration, cold, or heat stress conditions [15]. The expression level of the *ScDIR* gene increases under abiotic stresses such as Polyethylene glycol (PEG), NaCl, and H_2_O_2_ [37]. Most *VrDIR* genes in mungbean respond to drought and salt stress in different tissues [38]. Overexpression of the sugarcane *ScDIR7*, *ScDIR11*, *ScDIR40*, and *ScDIR5* genes in transgenic tobacco increases the lignin content, indicating that the sugarcane *ScDIR* genes are involved in lignin synthesis and plant drought stress resistance [39]. The loss of function of pepper *CaDIR7* reduces root activity under salt stress, and the induction of stress-related genes is inhibited in *CaDIR7*-silenced plants [40]. Furthermore, some *DIR* genes in plants such as *Phryma leptostachya* [41], *Medicago truncatula* [42], and *Pyrus bretschneideri* [43] also participate in various abiotic stresses. 

Understanding the role of stress genes in plant physiological processes will provide a viable avenue for analyzing and improving crop defense responses to biotic and abiotic stresses. As there are currently no literature reports on research involving the *SsDIR* gene family, this study intends to conduct a detailed and comprehensive analysis of the gene structure, conserved structural domains, phylogenetic tree, promoter cis-acting elements, collinearity relationships, etc., of the *SsDIR* family at the whole genome level through bioinformatics mining. In addition, *SsDIR* gene expression in different tissues and stress response gene patterns were analyzed to preliminarily explore the biological functions of the *DIR* gene family in *S. superba* and to provide a reference for the role of the *SsDIR* gene family in exploring the response process to biotic and abiotic stresses. 

## 2. Results

### 2.1. Identification and Physicochemical Properties of SsDIR Gene Family Members

Through HMM analysis and BlastP comparison, we initially identified 30 members of the *DIR* gene family in *S. superba*. Subsequently, after integrating conservative domain identification, a final selection of 24 *SsDIR* family members was achieved. These members were named *SsDIR1* to *SsDIR24* according to their positions on the chromosome. Using TBtools software, we predicted the physicochemical properties of the *SsDIR* proteins (Table 1). The results revealed that the number of amino acids in *SsDIR* proteins ranged from 154 to 512, with an average of 219.58. The amino acid count exhibited considerable variation. The relative molecular weight ranged from 16,885.29 to 55,676.08 Da, with an average of 20,985.3 Da. The isoelectric point ranged from 4.65 to 9.95, with an average of 6.49. Among them, there were 18 alkaline proteins (pI > 7) and 6 acidic proteins (pI < 7). The instability coefficient ranged from 14.37 to 42.97, with an average of 26.08. Among these, only 2 SsDIR proteins (SsDIR8 and SsDIR16) had instability coefficients greater than 40, indicating that they are unstable proteins, while the rest were considered stable proteins. In the 24 *SsDIR* gene family members, most *SsDIR* proteins exhibited a hydrophilic nature, with an average hydrophilicity value greater than 0, indicating that the majority of SsDIR proteins are hydrophobic. Only 4 proteins (SsDIR6, SsDIR10, SsDIR13, SsDIR14) had average hydrophilicity values less than zero, suggesting that they are hydrophilic proteins.

### 2.2. Phylogenetic Analysis and Classification of SsDIRs

Using the sequences of *DIR* proteins from *S. superba*, *A. thaliana*, *O. sativa*, and *P. trichocarpa*, we constructed a phylogenetic tree using the neighbor-joining (NJ) method. Based on the homology analysis of SsDIR proteins and referring to the classification of AtDIR proteins, we divided the SsDIR proteins into five subfamilies: DIR-a, DIR-b/d, DIR-c, DIR-e, and DIR-g (Figure 1). Among them, the AtDIR members are divided into three subfamilies: DIR-a, DIR-b/d, and DIR-e, containing 5, 14, and 6 genes, respectively. The distribution of DIR proteins in *S. superba* is similar to that in *A. thaliana*, with proteins distributed in the DIR-a, DIR-b/d, and DIR-e subfamilies. Among these, the DIR-b/d subfamily contains the highest number of SsDIR proteins, with 13 members, followed by the DIR-a subfamily with only 6 genes, and the DIR-e subfamily with the fewest members, comprising SsDIR2, SsDIR15, SsDIR16, SsDIR21, and SsDIR22. 

### 2.3. Conserved Motif, Conserved Structural Domains, and Gene Structure Analysis of SsDIRs

Analysis of the gene structure of the *SsDIR* family proteins revealed (Figure 2) that most *SsDIR* genes exhibit the classical *DIR* gene structure with a single exon and no introns. However, there are exceptions, with a few genes such as *SsDIR10*, *SsDIR13*, and *SsDIR14* containing two exons and one intron. Analysis of the conserved motifs of the *SsDIR* family proteins found that among 24 SsDIR proteins, 10 conserved motifs were identified, and all of the proteins had 3 to 6 conserved motifs. Among these, motif1, motif2, and motif3 were found in all three subfamilies, suggesting their widespread distribution and strong conservation within the SsDIR protein sequences. Other motifs only appeared in specific subfamilies: motif4, motif5, and motif6 are exclusively distributed in proteins of the *DIR*b/d subfamily, while motif7 is found only in the DIR-a subfamily. Motif8 and motif9 are present solely in proteins of the DIR-e subfamily. From the distribution pattern of conserved motifs in the SsDIR protein family, it is evident that proteins within the same subfamily exhibit similar arrangements of conserved motifs. Conversely, noticeable differences exist in the conserved motifs between different subfamilies. This observation underscores the functional diversity of *SsDIR* across various subfamilies.

### 2.4. Chromosome Distribution of SsDIRs

The chromosomal localization analysis of the *SsDIR* gene family revealed (Figure 3) that the 24 *SsDIR* family genes are unevenly distributed across 13 chromosomes. Chr6 harbors four *SsDIR* genes, while Chr5, Chr8, and Chr15 each contain three *SsDIR* genes. Chr16 and Chr17 both host two *SsDIR* genes, while the remaining chromosomes each carry one *SsDIR* gene. To analyze the selective pressure on the *SsDIR* gene family, the Ka, Ks, and Ka/Ks values of duplicated gene pairs were calculated. A total of four pairs of *SsDIR* genes, involving eight *SsDIR* genes, were obtained. The Ka/Ks values of the duplicated gene pairs in the *SsDIR* gene family ranged from 0.1061 to 0.3220, all of which were less than one (Appendix A), indicating that the duplicated gene pairs in the *S. superba DIR* gene family may have undergone purifying selection during evolution.

### 2.5. Intraspecific and Interspecific Collinearity Analyses

To investigate gene duplication events within the *SsDIR* gene family, a synteny analysis was conducted using TBtools software. The results revealed four pairs of gene duplicates (*SsDIR2/SsDIR16*, *SsDIR4/SsDIR12*, *SsDIR7/SsDIR11*, *SsDIR8/SsDIR20*) within collinear regions of the *S. superba* genome, distributed across seven chromosomes (Chr3, 5, 6, 7, 8, 13, 15), aggregating into four regions of segmental duplication events. The *SsDIR* gene family harbors a total of six tandem duplication sequences, involving the formation of *DIR* gene clusters on chromosomes (Figure 4a). 

To further explore the evolutionary relationships of the *SsDIR* genes, we also investigated the *DIR* gene family members in *S. superba* and other species, with *O. sativa* representing monocots and *A. thaliana* and *P. trichocarpa* representing dicots. The *SsDIR* genes were subjected to inter-species collinearity analysis with the genomes of *P. trichocarpa*, *A. thaliana*, and *O. sativa* (Figure 4b). The results revealed that *S. superba* shares the closest evolutionary relationship with the woody model plant, *P. trichocarpa*, exhibiting 24 pairs of collinear gene pairs. Following this, it showed 20 collinear gene pairs with the dicot model plant *A. thaliana*, and only 2 collinear gene pairs with the monocot model plant *O. sativa*. The collinear-related genes are distributed across chromosomes 1, 3, 4, 5, 6, 7, 8, 9, 10, 13, and 16. Both *SsDIR16* and *SsDIR2* are found in the *DIR* genes of *A. thaliana*, *O. sativa*, and *P. trichocarpa*. *DIR7*, *DIR11*, *DIR12*, and *DIR15* simultaneously exhibit collinearity with two distinct *A. thaliana* and *P. trichocarpa DIR* genes, indicating the presence of potential orthologous gene pairs. These genes may have played significant roles in the evolutionary process of the *DIR* gene family.

### 2.6. Analysis of Cis-Acting Elements in SsDIR Promoters 

Trans-acting elements play crucial roles in plant growth, stress responses, and tissue-specific expression. Extracting the upstream 2000 bp sequences from the transcription start sites of *SsDIR* genes, we analyzed the cis-acting elements in their promoters (Figure 5). The results indicate the presence of 6 stress-responsive elements, 26 light-responsive elements, 7 plant growth and development elements, and 10 plant hormone-responsive elements. In terms of plant hormone-responsive elements, *SsDIR* genes are enriched in abscisic acid-responsive elements (ABREs) and methyl jasmonate-responsive elements (CGTCA-motif and TGACG-motif). Furthermore, all *SsDIR* gene promoters contain both light-responsive and plant hormone-responsive elements, indicating that the expression of these members is regulated by light and involved in the regulation and metabolism of plant hormones. Furthermore, in the *SsDIR* family, stress-responsive elements are widely distributed, with all members except *SsDIR9* containing stress-responsive elements. These include the ARE (anaerobic induction response element), MBS (drought-responsive element), TC-rich repeats (stress-responsive element), LTR (low-temperature-responsive element), WUN-motif (wound-responsive element), and GC-motif. The members of the *SsDIR* family containing AREs are the most numerous, as shown in Appendix A. The significant presence of cis-elements also indicates that the *SsDIR* gene can respond to different stresses. 

### 2.7. Functional Interaction Network of SsDIR Proteins

To explore the interaction patterns of *DIR* genes in *S. superba*, we predicted the SsDIR protein interaction network based on *A. thaliana* DIR proteins (Figure 6). The SsDIR proteins form a network with 16 nodes, each interacting with multiple other nodes. For example, *SsDIR15*, *SsDIR7*, *SsDIR5*, *SsDIR16*, and *SsDIR9* have numerous connections to other nodes, indicating the diversity of DIR proteins. GO enrichment analysis showed that these proteins are primarily involved in the pinoresinol biosynthetic process (GO:1901599), lignan biosynthetic process (GO:0009807), phenylpropanoid biosynthetic process (GO:0009699), and guiding stereospecific synthesis activity (GO:0042349).

### 2.8. Expression Patterns of SsDIRs under Drought Conditions

We selected 11 genes from three different subfamilies, including *SsDIR4* from the DIR-a subfamily; *SsDIR1*, *SsDIR3*, *SsDIR7*, *SsDIR10*, *SsDIR20*, *SsDIR17*, and *SsDIR23* from the DIR-b/d subfamily; and *SsDIR15*, *SsDIR21*, and *SsDIR22* from the DIR-e subfamily, and analyzed their expression levels after drought treatment. As depicted in Figure 7 and Appendix A, the expression of these 11 genes varies under drought stress. For most genes (*SsDIR3*, *4*, *7*, *10*, *15*, *17*, *20*, *23*), their expression levels decrease with the increasing severity of drought stress and significantly increase after rehydration. Specifically, *SsDIR4* and *SsDIR17* show a noticeable increase in expression after drought and rehydration, indicating that drought stress may temporarily inhibit the expression of these genes. On the contrary, the expression levels of the *SsDIR1*, *SsDIR21*, and *SsDIR22* genes peak in the late stages of drought (Figure 8a), indicating that the expression levels of these three genes increase with the severity of drought stress, positively responding to drought stress. 

### 2.9. Expression Patterns of SsDIRs in Different Tissue Sites

To further analyze the expression patterns of *SsDIR* genes in different tissues, we selected 11 genes from three different subfamilies and analyzed their expression levels in six different tissues: roots, stems, leaves, bark, phloem and xylem. As shown in Figure 9 and Appendix A, members of the *SsDIR* family are expressed in all tissues, but their expression patterns vary significantly among different tissues. Specifically, the *SsDIR3* and *SsDIR17* genes exhibit similar expression patterns, with the highest expression levels in roots and the lowest expression levels in stems, leaves, and bark. On the other hand, the *SsDIR7* and *SsDIR10* genes show high expression levels in *S. superba* root, stem, leaf, phloem and xylem tissues but low expression levels in the bark tissue. In contrast, *SsDIR1*, *SsDIR21*, and *SsDIR22* were specifically expressed in the bark tissues and were expressed at lower levels in other sites. In addition, *SsDIR15* was relatively highly expressed in leaf, phloem, and xylem tissues, and *SsDIR4* was expressed in xylem tissue. The results indicate that most *SsDIR* genes exhibit the highest expression levels in root tissues (Figure 8b), while a few show the highest expression in leaf in bark tissues. Specifically, the *SsDIR3*, *SsDIR7*, *SsDIR10*, *SsDIR17*, *SsDIR20*, and *SsDIR23* genes are highly expressed in root tissues, while the *SsDIR1*, *SsDIR21*, and *SsDIR22* genes show the highest expression in leaf in bark tissues. These findings reveal the diverse expression patterns of the *SsDIR* gene family in different tissues of *S. superba*. 

## 3. Discussion

The *DIR* gene family is principally involved in the biosynthesis of lignans and lignins and exerts significant roles in plant growth, development, and responses to biotic and abiotic stresses. The dirigent protein is widely distributed in terrestrial plants. In model plants such as *A. thaliana* and *O. sativa*, 25 and 49 *DIR* genes have been identified, respectively. Subsequently, in *Brassica*, *Isatis indigotica*, and *P. bretschneideri*, 29, 19, and 35 *DIR* genes have been identified, respectively [6,14,43]. To date, there have been no reports on the study of the *DIR* gene family in *S. superba*. Therefore, this study is based on genomic data of *S. superba* obtained by our research group to identify and analyze the *DIR* gene family in *S. superba*. Furthermore, the expression patterns of *S. superba DIR* genes in different tissues and under abiotic stress (drought treatment) were analyzed using qRT-PCR. 

In this study, a total of 24 members of the *SsDIR* gene family were identified and characterized. Most of these genes encode stable proteins, exhibiting physicochemical properties similar to the *DIR* proteins found in *Capsicum annuum* L. [40], *Fragaria vesca* [36], and *S. melongena* [44]. The phylogenetic tree analysis classified the 24 members of the *SsDIR* gene family into three subfamilies: DIR-a, DIR-e, and *DIR*b/d. This classification is similar to the phylogenetic clustering results of the *DIR* gene families in *C. annuum* [40], *P. trichocarpa* [20], and *S. melongena* [44]. There are significant differences in the gene structures of the *DIR* genes in the three different subfamilies. The conserved motifs and gene structures of most members of the same subfamily are similar, indicating that genes in the same family have a close evolutionary relationship.

Based on research, members of the DIR-a subfamily play a role in lignin biosynthesis. For instance, *AtDIR6* and *PbDIR* have been shown to participate in lignin biosynthesis [16,43]. Therefore, it is possible that the DIR-a family members in *S. superba* (*SsDIR4*, *SsDIR5*, *SsDIR6*, *SsDIR12*, *SsDIR13*, *SsDIR14*) are also involved in lignin synthesis. However, the precise mechanisms of these members remain to be further investigated. Exploring the biological function of the *DIR* genes and elucidating the lignin biosynthesis pathway in *S. superba* will provide valuable insights. In the DIRb/d subfamily, which has the highest number of members, tandem repeat genes are predominantly present. This suggests a certain expansion trend for DIR-b/d subfamily genes. Similar results have been partially confirmed in plants such as *Gossypium hirsutum* [33], *Picea* spp. [7], and *Linum usitatissimum* L. [45]. Additionally, tandem duplications in plants are considered an appropriate response to the constantly changing environment [46]. For example, overexpression of soybean *GmDIR2* and *GhDIR1* leads to an increase in the lignin content, enhancing plant resistance to pests and diseases [33,34]. Therefore, subsequent experiments can investigate the gene functions of the 12 members of the *DIR*b/d subfamily in *S. superba*. 

The majority of *SsDIR* genes have only one exon and lack introns, which is consistent with the gene structures of *DIR* genes in other plants such as *P. bretschneideri* [43], *C. annuum*r [40], *P. trichocarpa* [20], and *S. melongena* [44]. However, in *O. sativa* [17], one-third of the *DIR* genes contain at least one intron, indicating that the evolutionary mechanisms of monocotyledonous and dicotyledonous plants may differ. Chromosomal localization analysis of *S. superba* revealed that the *SsDIR* gene family is distributed across 13 chromosomes, with six pairs of tandem duplication sequences and four pairs of segmental duplication gene pairs. This suggests that the expansion of the *SsDIR* gene family is the result of both tandem duplication and segmental duplication. Similar phenomena have been observed in gene families of *O. sativa* [17], *F. vesca* [36], and other plants. The Ka/Ks ratios of *SsDIR* gene pairs are all less than 0.1, indicating that the *SsDIR* gene family may have undergone strong purifying selection pressure during evolution [47]. This phenomenon is consistent with the results observed in *S. tuberosum* [22] and *P. trichocarpa* [20]. 

The collinearity analysis of the *DIR* gene family in *A. thaliana*, *O. sativa*, and *P. trichocarpa* revealed that *S. superba* shares 24 collinear gene pairs with *P. trichocarpa*, 2 collinear gene pairs with *O. sativa*, and 20 collinear gene pairs with *A. thaliana.* This indicates that the *DIR* family in *S. superba* exhibits more homologous similarity with the *DIR* genes in *A. thaliana* and *P. trichocarpa*, with a closer evolutionary relationship between *S. superba* and *P. trichocarpa*. The *DIR* genes show more homology and conservation among dicotyledonous plants, consistent with the findings in *S. tuberosum* [22]. 

In the analysis of cis-acting elements in the promoter region of the *SsDIR* gene family, it was identified that there are light-responsive elements and anaerobic, drought, and low-temperature responsive elements, indicating that the *SsDIR* protein plays an important role in both biotic and abiotic stress. Additionally, there are also plant hormone response elements such as MeJA, abscisic acid (ABA), and SA, indicating that the expression of *SsDIR* genes is influenced by multiple plant hormones. This is consistent with the findings of the promoter cis-acting element analysis in *F. vesca* [36]. Among them, a large number of *SsDIR* genes respond to the regulation and metabolism of MeJA and ABA. Previous studies have indicated that hormones such as MeJA and ABA can alleviate plant stress damage by increasing antioxidant enzyme activities (POD), scavenging free radicals, and increasing the content of osmotic regulatory substances [48,49,50,51]. Therefore, it is speculated that *SsDIR* may be involved in the regulation of abiotic stress by responding to hormones such as MeJA and ABA, thereby enhancing plant tolerance to adverse environmental conditions. However, the specific mechanism requires further investigation. 

We subjected *S. superba* to drought treatment and selected certain genes for qRT-PCR analysis. The experimental results showed that the expression levels of the *SsDIR3*, *SsDIR4*, *SsDIR10*, and *SsDIR23* genes significantly decreased with the duration of drought, almost to the point of no expression. In contrast, *SsDIR7*, *SsDIR15*, *SsDIR17*, and *SsDIR20* responded to drought at 96 h, but their expression levels still showed a downward trend. The expression levels of *SsDIR1*, *SsDIR22*, and *SsDIR23* significantly increased with the duration of drought, showing a strong response. After four days of drought, their expression levels increased nearly 30-fold, indicating a robust response to drought stress. Similarly, the upregulation of DIR family members has been observed in *Saccharum officinarum* L. [37] and *S. tuberosum* [22], suggesting a potential role of the *DIR* genes in drought stress tolerance mechanisms. However, further experiments are needed to elucidate the specific mechanisms involved. The differential expression of these 11 genes under drought stress indicates that SsDIRs are involved in the plant’s response to abiotic stress. The increased expression of *ScDIR* genes under abiotic stress indicates that *ScDIR* is involved in the response to abiotic stresses such as drought and salt [37,39]. *AtDIR5* changes under drought stress [26]; *VrDIR* expression increases under drought stress [38]; and *SiDIR19*, *SiDIR20*, *SiDIR22*, *SiDIR27*, and *SiDIR36* are upregulated after drought treatment, responding to drought stress [21]. Based on the research findings of the *DIR* gene family in various plants, it is hypothesized that the *DIR* gene family in *S. superba* may have gene functions related to abiotic stress responses.

The expression patterns of *DIR* genes in various tissue types exhibit heterogeneity across different species such as *L. usitatissimum* [45], *O. sativa*, [17] *Brassica* [6], and *C. annuum* [40]. In different parts of *S. superba* officinalis, the *SsDIR* gene expression differs: three genes (*SsDIR1*, *SsDIR21*, and *SsDIR22*) are significantly expressed in the bark, while six genes (*SsDIR3*, *SsDIR7*, *SsDIR10*, *SsDIR17*, *SsDIR20*, and *SsDIR23*) are highly expressed in the roots. The results indicate that half of the *SsDIR* genes are expressed in the roots, which aligns with findings in *A. thaliana* [26], where 60% of the *DIR* genes are expressed in the roots, and in *O. sativa* [52], *P. trichocarpa* [20], *S. officinarum* [37], *Brassica* [6], and *I. indigotica* [14], where most genes are expressed in the roots, consistent with our results. The reason for the predominant expression in roots may be attributed to the presence of the Casparian strip in plant roots, serving as a lignin-based diffusion barrier crucial for maintaining nutrient balance. In vascular plants, root permeability is governed by the endodermis and the Casparian strip, while the dirigent protein family regulates lignin synthesis in the Casparian strip [53,54]. The highly expressed genes in the root belong to the DIR-b/d subfamily, indicating the significant role of DIR-b/d subfamily members in root development.

Specifically, genes associated with drought stress, such as *SsDIR1*, *SsDIR21*, and *SsDIR22*, exhibit robust expression in the bark, thereby facilitating lignin synthesis in response to drought stress. The specific mechanisms of their expression in the bark may require further investigation. Additionally, *SsDIR4*, belonging to the DIR-a subfamily, shows high expression in the xylem, where this subfamily is known to play a crucial role in lignin or lignan biosynthesis. This suggests that *SsDIR4* may be involved in lignin or lignan formation, although its specific mechanism requires further investigation. 

In vascular plants, lignification of nutrient organs is crucial for healthy plant growth [55]. Examples such as the seed coat of *A. thaliana* [25], the pod wall of soybean [56], and the hypocotyl of hemp [57] exhibit evidence of *DIR* gene involvement in lignification. This highlights the significance of *DIR* genes for the robust growth of plants. The varying expression patterns of the *SsDIR* gene family across different tissues of *S. superba* suggest their potential involvement in tissue development or lignification processes during biotic and abiotic stress responses. 

## 4. Materials and Methods

### 4.1. Identification and Physicochemical Properties of SsDIR Gene Family Members

The genomic sequence data and annotation information of *S. superba* are derived from the unpublished sequencing data of our research group. First, download the Hidden Markov Model (HMM) file (PF03018) for the DIR domain from the Pfam database (http://pfam.xfam.org/search, accessed on 3 December 2023) [58]. Then, use the Simple HMM Search in TBtools to search the *S. superba* protein database and extract protein sequences containing the DIR domain (E-value < 1 × 10^−5^). Meanwhile, download the protein sequences of the *AtDIR* gene family from the TAIR website (https://www.arabidopsis.org/index.jsp, accessed on 3 December 2023). Perform a homology comparison using Blastp to align these sequences with the *S. superba* protein sequences (E-value < 1 × 10^−5^). Take the intersection of the results identified by the two methods as the preliminary candidate *S. superba* DIR genes. Then, verify the conserved domains of the candidate genes using NCBI CDD Search (E-value < 1 × 10^−5^) [59], and remove those with incomplete domains. Finally, determine the members of the *SsDIR* gene family. Rename the *SsDIR* gene family members based on their chromosomal distribution. Analyze their physicochemical properties using the Protein Parameter Calc function in TBtools (version 1.108) [60]. 

### 4.2. Phylogenetic Analysis and Classification of SsDIRs

To investigate the systematic evolutionary relationships between the SsDIRs and DIRs of other species, download the AtDIR protein sequences from the TAIR database (https://www.arabidopsis.org/index.jsp, accessed on 3 December 2023) and the *P. trichocarpa* and *O. sativa* DIR protein sequences from the Phytozome database (https://phytozome.jgi.doe.gov/pz/portal.html, accessed on 2 January 2024). Perform multiple sequence alignment using MUSCLE in MEGA 11 software and optimize the alignment results based on GeneDoc 3. Construct a phylogenetic tree of the DIR gene families of *S. superba*, *A. thaliana*, *O. sativa*, and *P. trichocarpa* using the neighbor-joining (NJ) method with a bootstrap value set to 1000 [61]. Finally, enhance the phylogenetic tree using the iTOL website (https://itol.embl.de/, accessed on 18 April 2024) [62]. 

### 4.3. Conserved Motif, Conserved Structural Domains, and Gene Structure Analysis of SsDIRs

Use the Visualize Gene Structure function of the TBtools software to visualize the gene structure diagram of *SsDIRs*. Utilize the NCBI CDD Search (https://www.ncbi.nlm.nih.gov/cdd/, accessed on 3 December 2023) to analyze the dirigent structural domain and signal peptide of *SsDIR* proteins, and then visualize it using the Visualize NCBI CDD Domain Pattern function of the TBtools software (version 1.108). Predict the motif sequences of *SsDIR* proteins using the online website MEME (https://meme-suite.org/meme/, accessed on 1 March 2024), set the maximum number of discovered conserved motifs to 10, and keep other corresponding parameters as default values. Utilize the TBtools software Simple MEME Wrapper to visualize the conservative motifs of *SsDIR* transcription factor proteins. Finally, visualize the gene structure, conservative domains, and motif analysis results of *SsDIRs* together using the TBtools software Gene Structure View (Advanced). 

### 4.4. Chromosome Distribution of SsDIRs

Retrieve the chromosomal location information of *SsDIRs* from the genome of *S. superba*, analyze the chromosomal gene density using the Gene Density Profile function of TBtools software, and ultimately generate a chromosome distribution map of *SsDIRs* using the Gene Location Visualize from the GTF/GFF tool in the TBtools software. Then, use TBtools to perform calculations and analyses of nonsynonymous (Ka) and synonymous (Ks) substitutions. Evaluate the selective pressure on homologous gene pairs during evolution using the Ka/Ks ratio: Ka/Ks > 1 indicates positive selection, Ka/Ks < 1 indicates purifying selection, and Ka/Ks = 1 indicates neutral selection.

### 4.5. Intraspecific and Interspecific Collinearity Analyses of SsDIRs

To investigate the collinearity relationships of *DIR* genes across different species, three species, *S. superba*, *A. thaliana*, and *P. trichocarpa*, were selected. Utilizing TBtools software, collinearity analysis was conducted within the *S. superba* species and across the *S. superba*, *A. thaliana*, *O. sativa*, and *P. trichocarpa* gene families, and the collinearity relationships of the *DIR* gene family within and among species were visualized. 

### 4.6. Analysis of Cis-Acting Elements in SsDIR Promoters 

Using GXF Sequences Extract in the TBtools software, the upstream 2000 bp nucleotide sequences of each member of the *SsDIR* gene family were extracted from the *S. superba* genome sequence to serve as promoter regions. Subsequently, the Plant CARE website (https://bioinformatics.psb.ugent.be/webtools/plantcare/html/, accessed on 20 April 2024) was employed to analyze the types, quantities, and functions of cis-regulatory elements within these promoter sequences [63]. 

### 4.7. Functional Interaction Network of SsDIR Proteins

We uploaded all SsDIR protein sequences to the STRING database (http://string-db.org, accessed on 22 April 2024) and selected homologous sequences from *A. thaliana* as a reference. After completing the BLAST step, we constructed a network using the highest-scoring genes (bitscore) with a confidence parameter of 0.700.

### 4.8. Stress Treatment, RNA Extraction, and qRT-PCR Analysis

This study used two-year-old *S. superba* seedlings and ten-year-old *S. superba* as experimental plant materials. To further clarify the response of SsDIR family members to drought stress, a drought stress experiment was conducted on *S. superba* in the laboratory incubator of the Guangdong Academy of Forestry. The plant materials selected for this experiment were two-year-old *S. superba* seedlings with consistent growth, cultivated at room temperature (65–70% humidity, 25 °C, 16/8 h day/night photoperiod) using light substrate soil. The two-year-old seedlings were grown and tested in a climate chamber with a relative humidity of 70%, 25 °C, 650 μmol m^−2^s^−1^ light intensity, and a 16/8 h day/night photoperiod. The experiment started with watering, followed by natural drought treatment at time points of 0 h, 24 h, 48 h, and 96 h. After 96 h of drought, the water content decreased to 25%, which reached extreme drought. Rewater treatments were then conducted at 1 h, 12 h, and 24 h. Each treatment was repeated three times, and *S. superba* leaves were collected at each time point for RNA extraction and qRT-PCR analysis. Additionally, different tissues of ten-year-old *S. superba* were selected, including roots, stems, leaves, bark, phloem, and xylem, for RNA extraction and qRT-PCR analysis. 

Subsequently, RNA extraction was performed on the collected *S. superba* leaf samples, followed by qRT-PCR analysis. Additionally, RNA extraction and qRT-PCR analysis were conducted on various tissue samples collected from five-year-old *S. superba* plants, including roots, stems, leaves, bark, phloem, and xylem, to investigate the gene expression profiles in different tissues. According to the instructions provided with the RNA extraction kit (FastPure Universal Plant Total RNA Isolation Kit), total RNA was extracted from various tissue samples of *S. superba*. Subsequently, the extracted RNA was reverse transcribed into cDNA using the HiScript III 1st Strand cDNA Synthesis Kit (+gDNA wiper), prior to conducting the qRT-PCR experiments. The actin gene was used as an internal control, and the qRT-PCR experiments were performed using the ChamQ Universal SYBR qRT-PCR Master Mix. The cycling parameters were set as follows: 95 °C for 30 s, then 95 °C for 10 s, and 60 °C for 30 s, for 40 cycles, and a melt cycle from 65 °C to 95 °C for 5 s. The reaction mixture was 20 µL, including 1 µL of cDNA, 0.4 µL of primer-F (10 µmol/L), 0.4 µL of primer-R (10 µmol/L), 10 µL of ChamQ SYBR^®^ qPCR Master Mix (Vazyme, Nanjing, China), and 8.2 µL of ddH_2_O. The relative gene expression was calculated using the 2^−ΔΔCt^ method. Primer design for the qRT-PCR experiments was conducted using Primer3 (https://www.primer3plus.com/index.html, accessed on 7 March 2024). The specific primer sequences used for the qRT-PCR are listed in Appendix A. The RNA extraction reagents and qRT-PCR reagents were sourced from Vazyme (Nanjing Vazyme Biotech Co., Ltd., Nanjing, China).

## 5. Conclusions

In this study, we identified 24 members of the *SsDIR* gene family based on *S. superba* genomic data. These were categorized into the three subfamilies DIR-a, DIR-b/d, and DIR-e and named according to their chromosomal positions. By employing bioinformatics methods, we analyzed the physicochemical properties, phylogenetic tree, cis-acting elements, gene structure, motif analysis, conserved domains, chromosome localization, and intra/inter-specific collinearity. qRT-PCR technology was utilized to analyze the stress response patterns and tissue-specific expression patterns. The results indicated that the *SsDIR* gene responded to drought stress and exhibited different expression patterns in various tissues, suggesting its diverse regulatory roles in the growth and development of *S. superba* plants. In addition, the *SsDIR1*, *SsDIR21*, and *SsDIR22* genes are rapidly induced under drought stress, suggesting that these genes collectively play a role in enhancing the drought tolerance of *S. superba*. Meanwhile, *SsDIR4* is highly expressed in the xylem, indicating its potential involvement in promoting lignin biosynthesis and increasing defense responses. This study provides comprehensive information on the *SsDIR* gene family and lays a theoretical foundation for further understanding the biological functions of the *DIR* genes in *S. superba.*

## Figures and Tables

**Figure 1 ijms-25-07467-f001:**
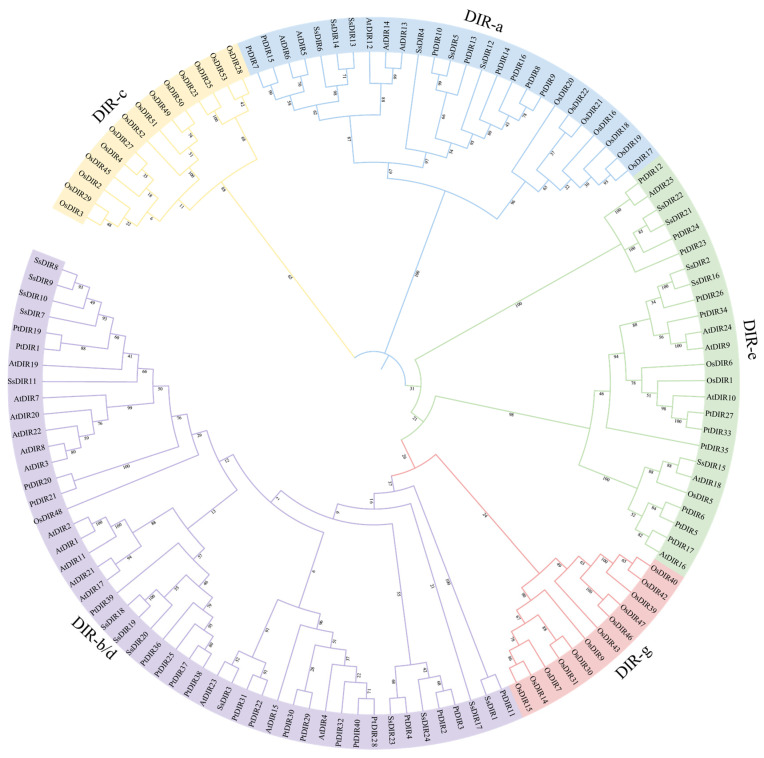
Phylogenetic analysis of *DIR* proteins from *S. superba*, *O. sativa*, *P. trichocarpa*, and *A. thaliana*. The tree was constructed using MEGA by the neighbor-joining method with 1000 bootstrap replicates.

**Figure 2 ijms-25-07467-f002:**
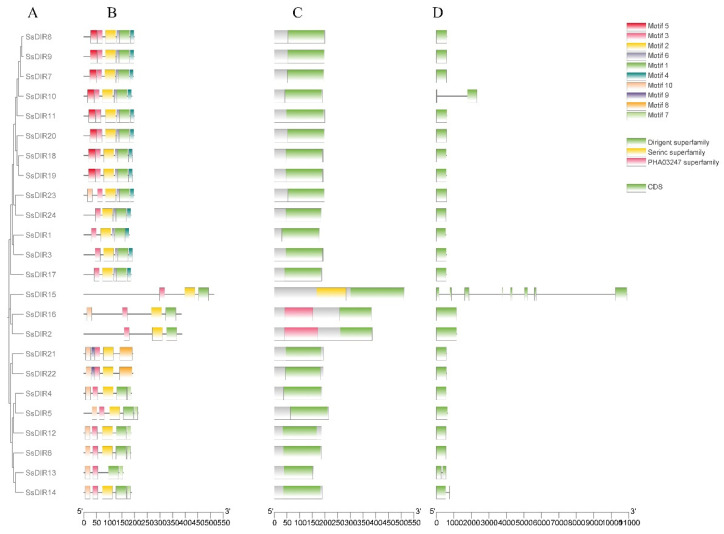
Phylogenetic relationship, conserved motif, and gene structure analyses of *SsDIRs*. (**A**) Phylogenetic tree of 24 *SsDIRs*. (**B**) Distribution of conserved motifs in *SsDIRs*. Ten putative motifs are shown in different colored boxes. (**C**) The conserved domains of *SsDIRs* were predicted and analyzed by NCBI-CDD. (**D**) Exon/intron organization of *SsDIRs*.

**Figure 3 ijms-25-07467-f003:**
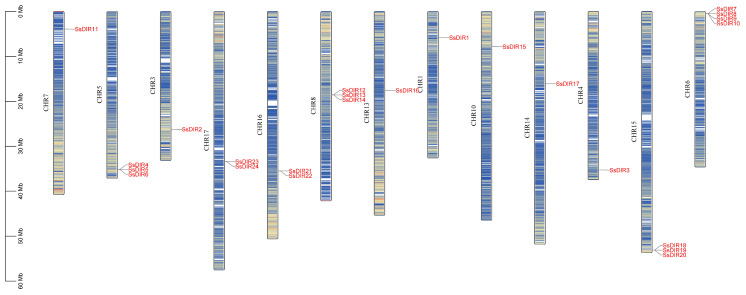
Chromosomal distribution of *SsDIR*s.

**Figure 4 ijms-25-07467-f004:**
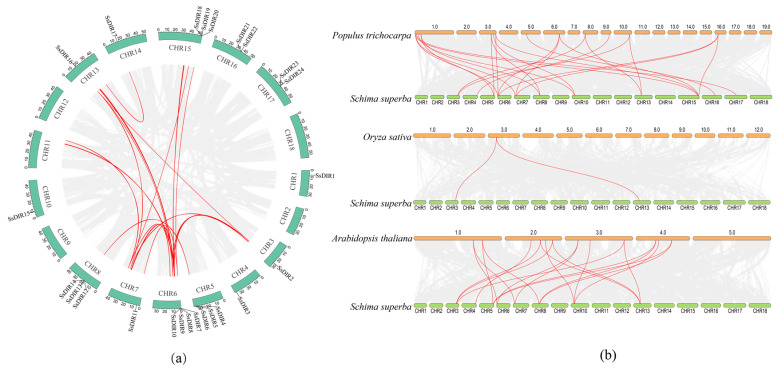
Synteny analysis of *SsDIR*s. (**a**) Schematic representations of the interchromosomal relationships of *SsDIR*s. (**b**) Synteny analysis of *DIR* genes among *S. superba*, *P. trichocarpa*, *A. thaliana*, and *O. sativa*. Gray lines in the background show the collinear blocks within *S. superba* and other plant genomes; the red lines highlight the syntenic *DIR* gene pairs.

**Figure 5 ijms-25-07467-f005:**
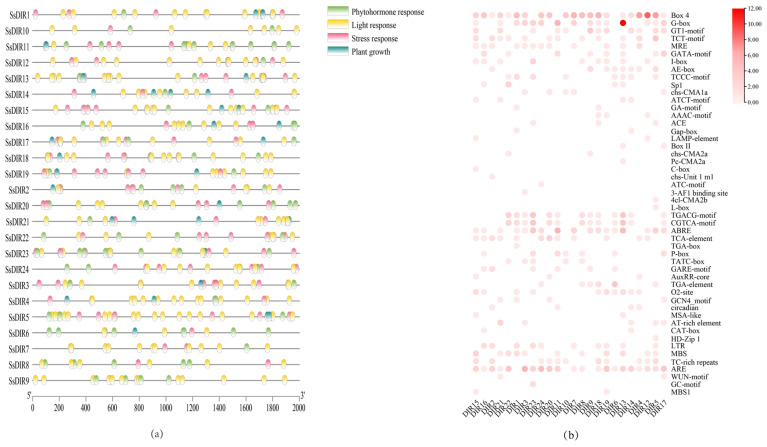
Predicted cis-elements in *SsDIR* promoters. The promoter sequences (−2000 bp) of 24 Ss*DIR*s were analyzed by PlantCARE. (**a**) Distribution of cis-acting elements in the promoter region. (**b**) Heat maps of cis-acting elementsfor the light-responsive, plant hormone-responsive, stress-responsive, plant growth and development, and the color concentration of the squares indicates the number of cis-acting elements.

**Figure 6 ijms-25-07467-f006:**
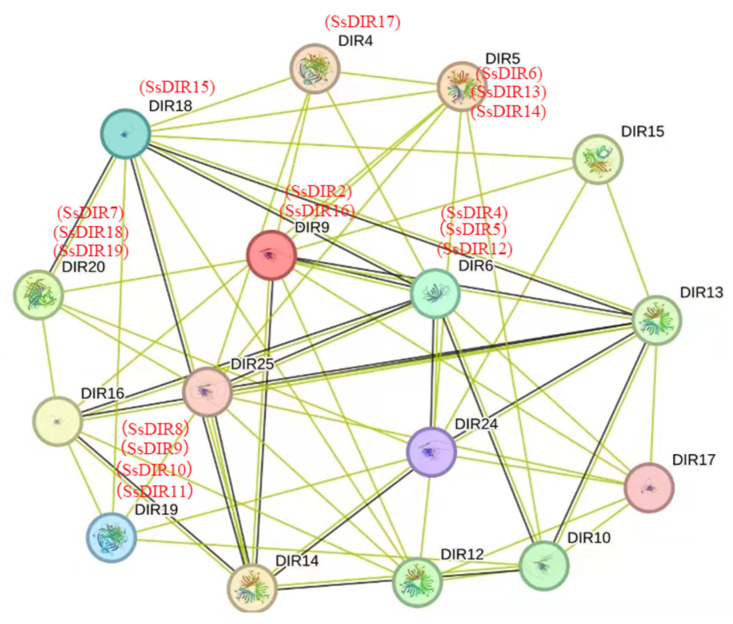
Prediction of the functional interaction network of *SsDIR* based on the orthologs in *A. thaliana*. The colored nodes: query proteins and first shell of interactors; green edges: predicted interactions with gene neighborhood; black edges: coexpression.

**Figure 7 ijms-25-07467-f007:**
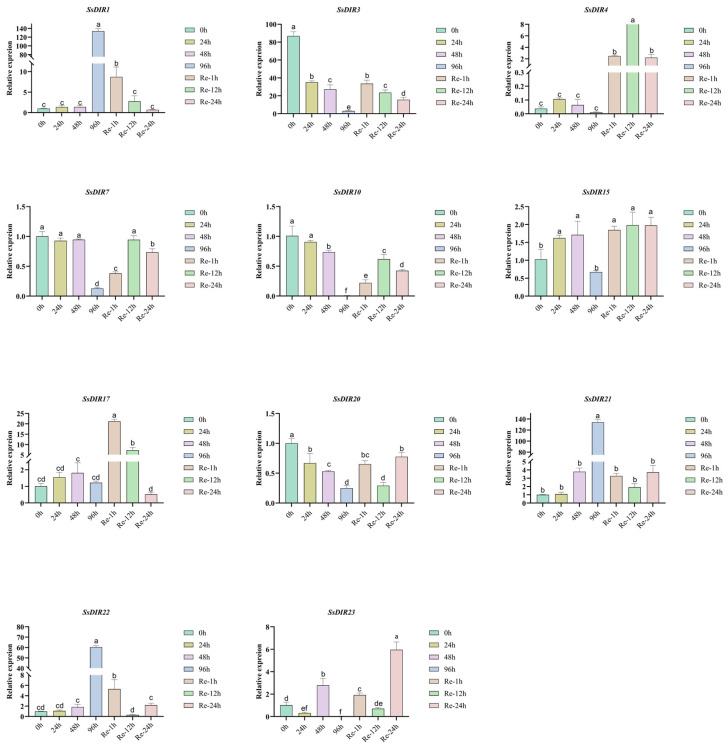
Relative expression levels of *SsDIR* genes under drought stress. The data were normalized with the expression level of 0 h by the 2^−ΔΔCt^ method. Error bars represent the mean ± standard deviation (SD) of three replications. Lowercase letters indicate significant differences at *p* < 0.05 according to ANOVA.

**Figure 8 ijms-25-07467-f008:**
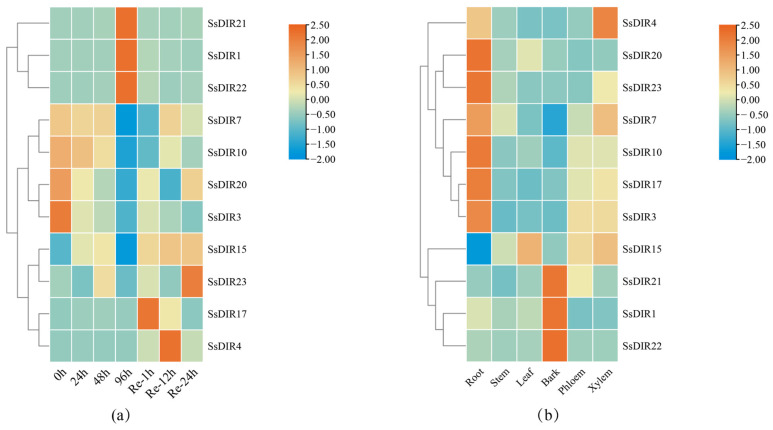
Expression profiles of *SsDIRs* in drought stress and different tissues. (**a**) Drought stress; (**b**) different tissues. The expression level was calculated according to the 2^−∆∆Ct^ method. The relative mRNA abundance of each gene was normalized with the *SsDIR* genes. The values indicate the means of three replications.

**Figure 9 ijms-25-07467-f009:**
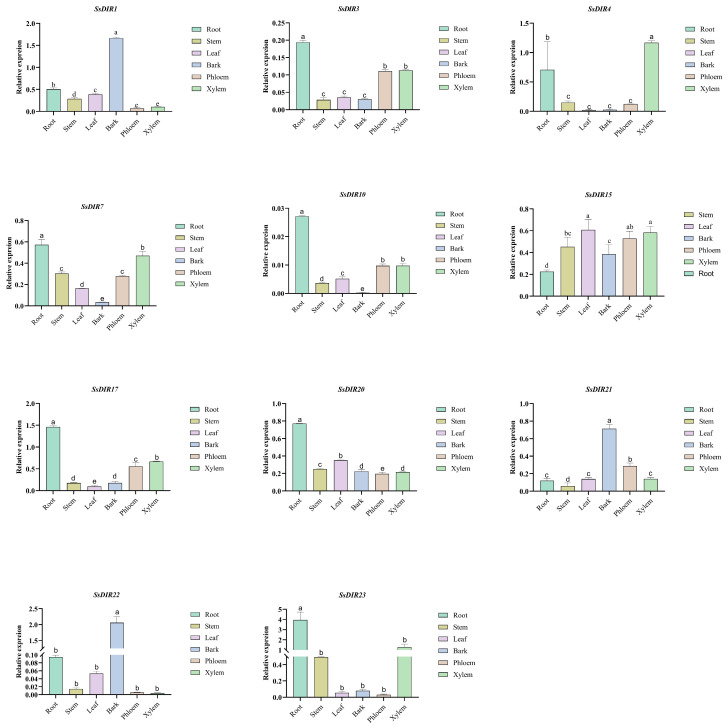
Relative expression of *SsDIRs* in different tissues. The data were normalized with the expression level of 0 h by the 2^−ΔΔCt^ method. Error bars represent the mean ± standard deviation (SD) of three replications. Lowercase letters indicate significant differences at *p* < 0.05 according to ANOVA.

**Table 1 ijms-25-07467-t001:** The physiochemical characteristics of the DIR gene family in *S. superba*.

Sequence ID	Number of Amino Acids	Molecular Weight	Theoretical pI	Instability Index	Aliphatic Index	Grand Average of Hydropathicity
*SsDIR*1	178	19,419.48	9.37	38.71	83.82	0.17
*SsDIR*2	386	39,000.97	4.66	32.94	95.44	0.191
*SsDIR*3	191	20,751.86	8.96	22.06	95.45	0.132
*SsDIR*4	187	21,162.19	5.82	37.6	80.32	0.012
*SsDIR*5	213	23,490.15	8.81	30.06	87.04	0.263
*SsDIR*6	184	20,653.67	7.7	26.23	66.79	−0.171
*SsDIR*7	195	21,434.72	8.86	34.4	91.44	0.085
*SsDIR*8	198	21,986.55	9.41	42.97	85.61	0.097
*SsDIR*9	197	21,859.35	9.55	38.92	83.1	0.112
*SsDIR*10	189	20,924.13	9.16	24.36	84.55	−0.06
*SsDIR*11	198	21,673.08	9.79	29.52	82.73	0.031
*SsDIR*12	185	21,055.11	6.96	23.88	80.7	0.045
*SsDIR*13	154	16,885.29	5.86	14.37	74.03	−0.05
*SsDIR*14	188	20,869.8	8.5	16.18	68.46	−0.177
*SsDIR*15	512	55,676.08	7.92	39.76	89.12	0.086
*SsDIR*16	383	39,307.03	4.65	40.46	86.29	0.021
*SsDIR*17	186	20,578.76	8.69	28.38	104.73	0.174
*SsDIR*18	191	21,064.42	9.79	23.25	87.85	0.093
*SsDIR*19	191	21,150.55	9.95	25.2	85.81	0.117
*SsDIR*20	197	21,600	9.89	32.48	88.63	0.06
*SsDIR*21	193	21,117.1	7.94	29.08	93.89	0.034
*SsDIR*22	192	20,950.98	7.95	26.46	96.88	0.116
*SsDIR*23	197	21,676.61	5.45	28.46	80.71	0.049
*SsDIR*24	185	20,387.54	9.26	35.3	77.03	0.057

## Data Availability

Data are contained within the article and Appendix A.

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
