# Peer review of "Genome-Wide Identification, Evolution, and Expression Analysis of the DIR Gene Family in Schima superba"

_ijms, 2024, doi:10.3390/ijms25137467_

Round 1

Reviewer 1 Report

Comments and Suggestions for Authors

Dear Editor,

Greetings,

I would like to thank you for giving such opportunity for reviewing the MS “Genome-Wide Identification, Evolution and Expression Analysis of the DIR Gene Family in Schima superba”, an important research area to develop S. superba.  The MS covered all in silico analysis, and gene expression analysis. Still, it needs more improvisation in terms of presentation.

1.      Authors need to re write the abstract with major findings.

2.      The introduction needs to reorganize and explain the importance of DIRs, their function, molecular mechanism, and involvement in different pathways.

3.      Authors need to include details about of drought stress in methods.

4.      Authors need to re check the gene expression levels. Some of them exhibiting greater than 1000-fold expression. Expand the results of qRT expression analysis results. Explain the gene expression results in folds instead of upregulation or down regulation / higher expression or decreased expression.

5.      Construct heat map for qRT expression analysis.

6.      Authors can perform protein interaction networks and discuss their functional roles.

7.      Authors can discuss the conserved domains with meme results, calculate evolutionary relationship ratios (ka/ks), and compare the exon-introns with evolutionary existence of genes in different crops.

8.      Authors need to discuss more about drought tolerance expression patterns of DIRs.

9.      Authors can attach primers list and qRT expression values as supplementary tables.

Author Response

Dear reviewer,

We would like to thank you for your professional review work, constructive comments, and valuable suggestions on our manuscript. We have carefully revised the manuscript and provided the point-by-point response. The changes in the revised manuscript have been highlighted. We hope these changes will strengthen our manuscript.

  • Authors need to re write the abstract with major findings.

Response: Thank you for your suggestion. We have re-written this part according to the Reviewer's suggestion(Line 10-26).

  • The introduction needs to reorganize and explain the importance of DIRs, their function, molecular mechanism, and involvement in different pathways.

Response: As suggested by the reviewer, we have added more references to explain the section on the DIR gene family(Line 71-109).

  • Authors need to include details about of drought stress in methods.

Response: In the drought experiment, Two-year-old Schima superba seedlings were cultivated and tested in a climate chamber with a relative humidity of 70%, 25°C, 650 μmol m⁻²s⁻¹ light intensity, and a 16/8h day/night photoperiod. At the start of the experiment, the seedlings were watered, and then subjected to natural drought treatment. The drought treatment time points were 0h, 24h, 48h, and 96h. After 96 hours of drought, the water content decreased to 25%, which reached extreme drought, and rewater treatment was carried out, and the rehydration time points were 1h, 12h and 24h. Each treatment was replicated three times, and Schima superba leaves were collected at each time point. This section has been detailed in section 4.8 of the Materials and Methods(Line 496-512).

  • Authors need to re check the gene expression levels. Some of them exhibiting greater than 1000-fold expression. Expand the results of qRT expression analysis results. Explain the gene expression results in folds instead of upregulation or down regulation / higher expression or decreased expression.

Response: Thank you for your valuable suggestion. I apologize for the oversight. Upon careful review, we found an error in the control group used for the calculations, which led to incorrect results. We have recalculated the qRT-PCR values and corrected them in the revised manuscript. Additionally, we have reanalyzed the results and discussion (Line 379-414).

  • Construct heat map for qRT expression analysis.

Response: We think this is an excellent suggestion, which has been added in the results section (Figure 9).

  • Authors can perform protein interaction networks and discuss their functional roles.

Response: Thank you for your suggestion. We have incorporated it into the article in lines 246-256 and 491-495.

  • Authors can discuss the conserved domains with meme results, calculate evolutionary relationship ratios (ka/ks), and compare the exon-introns with evolutionary existence of genes in different crops.

Response: Thank you for your suggestion. We have incorporated it into the article in lines 186-191 and 352-356.

  • Authors need to discuss more about drought tolerance expression patterns of DIRs.

Response: We sincerely appreciate the valuable comments.We have supplemented the discussion section with results on the expression patterns of DIRs under drought tolerance, and have cited relevant literature (Line 379-399).

  • Authors can attach primers list and qRT expression values as supplementary tables.

Response: Thank you for your suggestion. We will upload a supplementary material(Table S3, S4).

We would love to thank you for allowing us to resubmit a revised copy of the manuscript and we highly appreciate your time and consideration.

Sincerely,

Reviewer 2 Report

Comments and Suggestions for Authors

The article entitled “Genome-Wide Identification, Evolution and Expression Analysis of the DIR Gene Family in Schima superbacovered the various parameters of the DIR gene family of Schima superba including both in-silico and expression studies. Authors have done genome-wide identification, gene and protein structure analyses, phylogenetic analyses, promoter analyses, Collinearity analysis of the DIR gene family in the Schima superba. Moreover, the authors have performed the qRT-PCR experiment on 11 selected genes in 6 different tissues and drought conditions. The study depicted that these genes might be involved in the lignin biosynthesis and abiotic stress tolerance in Schima superba. However, all the results are based on the routine in silico gene family analysis, which could be supported by functional analysis to make it more novel and of scientific importance. Moreover, the present manuscript has various concerns, some of which are listed here.

1.     The introduction part lacks citations in many places, therefore suggested to add.

2.     Authors should maintain uniformity in plant names, either botanical names or common names.

3.     Write the species name of plant “Populus.

4.     Write the full form of PEG, JA, SA.

5.     Authors are suggested to avoid the repetitions of the discussion part in the results section eg. (Line-110-113, 186-187, 193-196).

6.     On what basis, the authors have selected the 11 genes from the total of 24 genes for expression analysis?

7.     In-silico sub-cellular localization analyses predicted the discrete results, authors could validate the localization of selected SsDIRs by sub-cellular localization assay.  

8.     The authors have depicted the MW values in Dalton, while in the table and text, it is mentioned as kDa.

9.     The bootstrap values should be included in the phylogenetic tree figure.

10.  The discussion part lacks sufficient information, authors are suggested to discuss the results properly.

11.  What tissue is used as a control in the case of expression analysis of SsDIRs in different tissue sites?

12.  The tool name should be MEGA (Line number: 359).

13.  The 4.1 and 4.2 sections under the Materials and Methods part are very casually written therefore suggested to improve.

14.  What conditions or methods are used for drought treatment? also needs to be included in the materials and method section.

15.  There are many typographical and grammatical errors throughout the manuscript which need to be corrected after thorough reading.

Comments on the Quality of English Language

The English language improvement is required.

Author Response

Dear reviewer,

We would like to thank you for your professional review work, constructive comments, and valuable suggestions on our manuscript. We have carefully revised the manuscript and provided the point-by-point response. The changes in the revised manuscript have been highlighted. We hope these changes will strengthen our manuscript.

  • The introduction part lacks citations in many places, therefore suggested to add.

Response: We sincerely appreciate the valuable comments. We have checked the literature carefully and added more references on Abiotic stress and DIR gene into the  introduction part in the revised manuscript (Line 71-108).

  • Authors should maintain uniformity in plant names, either botanical names or common names.

Response:We feel sorry for our carelessness. in our resubmitted manuscript, the plant name has been modified. Thanks for your correction.

  • Write the species name of plant “Populus”.

Response: Populus refers to Populus trichocarpa. This information has already been included in the text.

  • Write the full form of PEG, JA, SA.

Response: Thank you for your feedback. We have added this information in line 95 and 100 of the text.

  • Authors are suggested to avoid the repetitions of the discussion part in the results section eg. (Line110-113, 186-187, 193-196).

Response: We sincerely appreciate the valuable comments. We have re-written this part according to the Reviewer's suggestion.

  • On what basis, the authors have selected the 11 genes from the total of 24 genes for expression analysis?

Response: Thank you for your insightful comments. Based on the literature review, we selected genes that have been reported to show significant expression under similar conditions. Additionally, we considered selecting genes from different subfamilies to ensure a representative analysis of the gene family. After taking into account factors such as specific primers, we ultimately identified these 11 genes.

  • In-silico sub-cellular localization analyses predicted the discrete results, authors could validate the localization of selected SsDIRs by sub-cellular localization assay.

Response: Thank you for your valuable suggestion. We recognize the importance of subcellular localization experiments in the study of gene families. However, due to the large number of SsDIR gene family members, the extensive workload, and the tight timeline, we are currently unable to complete this part of the experiment.   Therefore, we have decided to remove the sections related to subcellular localization from the manuscript. The core focus of this study is to explore the response of the SsDIR gene family to drought stress, aiming to understand whether it responds to abiotic stress. We plan to further investigate subcellular localization experiments in future research. Meanwhile, we have emphasized other key aspects of the DIR gene family in the current manuscript.

  • The authors have depicted the MW values in Dalton, while in the table and text, it is mentioned as kDa.

Response: We feel sorry for our carelessness. in our resubmitted manuscript,we have corrected the unit from kDa to Da.(Line 129 )

  • The bootstrap values should be included in the phylogenetic tree figure.

Response: Thank you for your suggestion. We have already made modifications to the phylogenetic tree figure.(Line 156)

  • The discussion part lacks sufficient information, authors are suggested to discuss the results properly.

Response: Thank you for your suggestion. We have rediscussed the results and added more references to support this idea.(Line 327-339, 353-357 and 379-414)

  • What tissue is used as a control in the case of expression analysis of SsDIRs in different tissue sites?

Response: In this study, a mixed sample of six parts of Schima superba roots, stems, leaves, bark, phloem, and xylem was used as the control.

  • The tool name should be MEGA (Line number: 359).

Response: We sincerely thank the reviewer for careful reading as suggested by the reviewer, We have corrected it to MEGA.(Line number: 452)

  • The 4.1 and 4.2 sections under the Materials and Methods part are very casually written therefore suggested to improve.

Response: We have re-written this part according to the Reviewer's suggestion and added many details.(Line 429-456)

  • What conditions or methods are used for drought treatment? also needs to be included in the materials and method section.

Response: In the drought experiment, Two-year-old Schima superba seedlings were cultivated and tested in a climate chamber with a relative humidity of 70%, 25°C, 650 μmol m⁻²s⁻¹ light intensity, and a 16/8h day/night photoperiod. At the start of the experiment, the seedlings were watered, and then subjected to natural drought treatment. The drought treatment time points were 0h, 24h, 48h, and 96h. After 96 hours of drought, the water content decreased to 25%, which reached extreme drought, and rewater treatment was carried out, and the rehydration time points were 1h, 12h and 24h. Each treatment was replicated three times, and Schima superba leaves were collected at each time point. This section has been detailed in section 4.8 of the Materials and Methods.(Line 496-512)

  • There are many typographical and grammatical errors throughout the manuscript which need to be corrected after thorough reading.

Response: We tried our best to improve the manuscript and made some changes to the manuscript. These changes will not influence the content and framework of the paper. And here we did not list the changes but marked in red in the revised paper.

We would love to thank you for allowing us to resubmit a revised copy of the manuscript and we highly appreciate your time and consideration.

Sincerely,

Reviewer 3 Report

Comments and Suggestions for Authors

Although the manuscript has a great potential, I am surprised by the lack of methodological details. Without these, it is impossible to understand what was done and if any biased has occurred. Specifically:

- How many biological samples were used? From which tissues?

 - The authors detail the bioinformatic programs used but not the specific inputs or parameters used. However, results are highly dependent on this.

- rt-PCTs were performed, but how many replicates were performed? Where are the primer sequences used?

- Figure resolution need to be enhanced as the text is not readable. 

- Data availability: "Data are contained within the article and Supplementary Materials." - the raw data is not shown. Where are the supplementary materials?

I am sorry for not being more positive at this point, but without these I cannot perform an accurate review. Thus, I hope the authors can fix this.

Comments on the Quality of English Language

English is fine but several sentences are awkward. As if they were written by an AI tool. See for instance the first section of M&Ms.

Author Response

Dear reviewer,

We would like to thank you for your professional review work, constructive comments, and valuable suggestions on our manuscript. We have carefully revised the manuscript and provided the point-by-point response. The changes in the revised manuscript have been highlighted. We hope these changes will strengthen our manuscript.

  • How many biological samples were used? From which tissues?

Response: In the drought qRT-PCR experiment, we selected three Schima superba seedlings for treatment, including seven time points (0, 24, 48, 96 hours of drought, and 1, 12, 24 hours of rehydration). Samples were taken from the leaves of Schima superba at these seven time points.

In the qRT-PCR experiment for different parts, we used biological samples from three independent replicates to ensure the reliability and reproducibility of our results. Samples were taken from six different tissues of Schima superba: roots, stems, leaves, bark, phloem, and xylem.

  • The authors detail the bioinformatic programs used but not the specific inputs or parameters used. However, results are highly dependent on this.

Response: Thank you for your reminder. We have added this information to the Materials and Methods section.

  • RT-PCRs were performed, but how many replicates were performed? Where are the primer sequences used?

Response: The qRT-PCR experiments were conducted with three biological replicates and three technical replicates. For the biological replicates, samples from three individual plants were pooled together to form a composite sample. The primer sequences will be uploaded to the supplementary materials (Table S3).

  • Figure resolution need to be enhanced as the text is not readable. 

Response: We will revise the figure to enhance readability, and we appreciate the valuable suggestion.

  • Data availability: "Data are contained within the article and Supplementary Materials." - the raw data is not shown. Where are the supplementary materials?

Response: We feel sorry for our carelessness. In our resubmitted manuscript, we upload a supplementary material

We tried our best to improve the manuscript and made some changes in the manuscript. These changes will not influence the content and framework of the paper. And here we did not list the changes but marked in red in revised paper. We appreciate for reviewer’s warm work earnestly, and hope that the correction will meet with approval. Once again, thank you very much for your comments and suggestions.

Sincerely,

Round 2

Reviewer 1 Report

Comments and Suggestions for Authors

Dear Editor,

Greetings,

The authors addressed all my concerns. The MS can be accepted in its present form.

Thank you,

Author Response

Dear Reviewer,

We are deeply grateful for your review and valuable comments on our manuscript. We are delighted to receive the notification that our paper has been accepted. This achievement would not have been possible without the time and effort you invested during the review process.

Your detailed feedback and suggestions were crucial in improving the quality of our manuscript. With your guidance, we were able to better articulate our research findings and enhance the scientific rigor and readability of the paper.

Once again, thank you for your support and assistance with our work.

Yours sincerely,

Changya Chen

Email: 135779@163.com

Reviewer 2 Report

Comments and Suggestions for Authors

In the revised version of the manuscript, the authors have tried to incorporate all the suggested changes or answer the queries. However, some suggestions are still not made, which should also be done in the manuscript.

  1. Authors should maintain uniformity in plant names, either botanical names or common names.

2.     Authors should provide the full form of JA and SA at the place of their first use.

  1. Authors are suggested to avoid the repetitions of the discussion part in the results section eg. (Line 144-148, 231-233).

Comments on the Quality of English Language

Minor grammatical/English improvements can be made.

Author Response

Dear reviewer,

Thank you very much for taking the time to review our manuscript. We greatly appreciate your comments and suggestions, which have been invaluable in improving our paper. We have addressed each of your suggestions and provided corresponding responses and explanations for the revisions. The changes made based on your feedback are highlighted in the manuscript.

  1. Authors should maintain uniformity in plant names, either botanical names or common names.

Response: We sincerely appreciate your valuable suggestions. We apologize for our oversight, and after careful review, we have corrected the plant names.

  1. Authors should provide the full form of JA and SA at the place of their first use.

Response: Thank you for your feedback. We have added the necessary information in line 82 of the manuscript.

  1. Authors are suggested to avoid the repetitions of the discussion part in the results section eg. (Line 144-148, 231-233).

Response: Thank you very much for your suggestions. Following the reviewer’s advice, we have rewritten this section of the results and removed the redundant parts(Line-136-145, Line-227-230). We have also checked and revised other parts accordingly.

We would love to thank you for allowing us to resubmit a revised copy of the manuscript and we highly appreciate your time and consideration.

Sincerely,

Reviewer 3 Report

Comments and Suggestions for Authors

The authors have addressed all previous concerns/comments. I congratulate the authors for this study!

Author Response

(The authors gave the same response as above.)
